# Avalanche Susceptibility Mapping by Investigating Spatiotemporal Characteristics of Snow Cover Based on Remote Sensing Imagery along the Pemo Highway—A Critical Transportation Road in Tibet, China

**Ning Xi and Gang Mei \***

School of Engineering and Technology, China University of Geosciences (Beijing), Beijing 100083, China; ning@email.cugb.edu.cn
* Correspondence: gang.mei@cugb.edu.cn

**Abstract:** The Pemo Highway is a critical transportation road to Medog County in the Tibet Plateau (TP). Since its completion in 2021, the Pemo Highway has been prone to frequent avalanches due to heavy rainfall and snowfall. Despite the lack of monitoring stations along the highway and limited research conducted in this area, remote sensing imagery provides valuable data for investigating avalanche hazards along the highway. In this paper, we first investigated the spatiotemporal characteristics of snow cover along the Pemo Highway over the past two years based on the GEE platform. Second, we integrated snow, topography, meteorology, and vegetation factors to assess avalanche susceptibility in January, February, and March 2023 along the highway using the AHP method. The results reveal that the exit of the Duoshungla Tunnel is particularly susceptible to avalanches during the winter months, specifically from January to March, with a significant risk observed in March. Approximately 3.7 km in the direction of the tunnel exit to Lager is prone to avalanche hazards during this period. The recent "1.17 avalanche" event along the Pemo Highway validates the accuracy of our analysis. The findings of this paper provide timely guidance for implementing effective avalanche prevention measures on the Pemo Highway.

**Keywords:** Pemo Highway; avalanche susceptibility; snow cover; Sentinel-2; Google Earth Engine (GEE)





## 1. Introduction

Medog County, located in the southeastern Tibet Plateau (TP), is renowned for its ecological preservation and was the last county in China to gain road access. The completion of the Pemo Highway has transformed Medog County from an isolated "island" in winter to a region with improved connectivity. The Pemo Highway plays a vital role in supporting the economic and social development of Medog County, as well as facilitating border protection in the TP.

The Pemo Highway is situated in the Duoxiong River basin, traversing the Himalayan mountain range with an elevation difference of approximately 3000 m. Local news reports have documented multiple avalanche disasters along the Pemo Highway since 2015 [1,2], resulting in casualties. With urbanization extending into mountainous regions and an increasing number of tourists visiting Medog County, the transportation route faces heightened exposure to severe avalanche hazards, intensifying the conflict between avalanche disasters and human interests. However, the Pemo Highway first opened to traffic in 2021 and has not attracted much international attention, and few researchers have investigated avalanche hazards along the highway. Therefore, it is necessary to investigate the spatiotemporal characteristics of snow cover along the Pemo Highway since its construction and assess its avalanche susceptibility to enhance public awareness of avalanche disasters.

Currently, in snow cover identification research, remote sensing technology has made significant strides over the past few decades, emerging as the principal approach for large-scale and high-precision snow monitoring [3,4]. Optical remote sensing data such as Landsat [5], Sentinel-2 [6], and MODIS [7] boast a high degree of accuracy in snow identification and have become indispensable in snow research endeavors [8,9]. Common methods for snow identification using optical remote sensing data include the normalized difference snow index (NDSI) [10,11], band threshold segmentation, and image classification. NDSI, effectively identifying snow based on its reflectance properties in the visible spectrum, has been widely adopted. For example, Tong et al. [12] investigated the sensitivity of NDSI thresholds in snow cover mapping and demonstrated its remarkable accuracy in snow extraction.

Historical studies have indicated that snow is a key factor contributing to avalanche occurrences along the Pemo Highway [13–16]. Therefore, it is of great significance to assess avalanche susceptibility with snow as the main conditioning factor after obtaining the spatial and temporal characteristics of the snow cover along the Pemo Highway. However, winter snow along the Pemo Highway primarily accumulates in remote mountainous areas, characterized by harsh environments and limited accessibility, which poses challenges for establishing meteorological and ground monitoring stations along the highway. The limited availability of data remains a primary challenge for avalanche susceptibility assessment along the Pemo Highway.

The introduction of Google Earth Engine (GEE) in 2015, a cloud computing platform for remote sensing, has revolutionized the analysis of massive amounts of geospatial data and facilitated online processing of remote sensing data. GEE has been extensively utilized in addressing various social issues [17–19], including forest degradation, disaster early warning, and environmental protection. The abundant optical remote sensing data stored in GEE, coupled with real-time meteorological data and its powerful online processing capabilities, offer a promising solution to assess avalanche susceptibility along the Pemo Highway, where meteorological data are scarce and accessibility is limited.

In recent avalanche susceptibility assessment research, most studies tend to utilize data-driven statistical methods and machine learning approaches to determine the significance of various conditioning factors, such as snow, terrain, meteorology, and land cover. This aids in the establishment of an objective evaluation system for avalanche susceptibility. The analytic hierarchy process (AHP) serves as a typical multi-criteria decision analysis (MCDA) [20,21] method frequently employed in statistics-based avalanche susceptibility research [13,20,22]. It prioritizes the conditioning factors based on specific criteria. In recent years, various types of efficient machine learning algorithms have emerged [23–25] and have been increasingly used in avalanche research. Machine-learning-based avalanche susceptibility analysis entails inputting a set of avalanche occurrence locations and conditioning factor data into various supervised learning models to train complex nonlinear relationships between avalanche disasters and their conditioning factors. Common methods encompass support vector machine (SVM) [26], random forest (RF) [27], k-nearest neighbor (KNN) [28], and classification and regression tree (CART) [29]. However, machine-learning-based methods often necessitate extensive training data to determine the significance of conditioning factors. Under data limitations, AHP has proven to be an effective solution for investigating avalanche susceptibility along the Pemo Highway.

Based on the unique characteristics of the Pemo Highway and the current state of avalanche research, in this paper, we first investigate the spatiotemporal characteristics of snow cover along the Pemo Highway over the past two years based on the GEE platform. Second, we attempt to employ the NDSI instead of the snow depth commonly used in past studies [28,30], and then integrate the topographic, meteorological, and vegetation factors to assess the avalanche susceptibility along the Pemo Highway from January to March 2023 based on the AHP method, highlighting the most hazardous section of the entire highway. Finally, we discuss the latest avalanche event that occurred in January 2023 on the Pemo Highway. Given the critical role of the Pemo Highway in the economic development

of Medog County, our research work can serve as a valuable reference for implementing effective avalanche protection measures along the highway. We encourage further studies by scholars to contribute to the economic progress and safety of the people in Medog.

The remainder of the paper is organized as follows. Section 2 introduces the study area and the data used in this paper. Section 3 introduces the spatial and temporal characteristics of snow cover along the Pemo highway. Section 4 introduces the avalanche susceptibility along the Pemo Highway. Section 5 discusses the latest "1.17 avalanche" in detail and the limitations of this paper. Section 6 concludes this paper.

## 2. Study Area and Data

### 2.1. Study Area

Medog County, situated in southeastern TP, China, is characterized by its low altitude, mild climate, abundant rainfall, and well-preserved ecology, making it a unique region in TP. However, the area faces challenges due to its complex topography, diverse geology, and special climatic conditions, leading to frequent geological hazards and heavy rainfall. These factors have resulted in significant transportation difficulties, impacting the daily lives of local residents, hindering tourism, and impeding the economic development of Medog County.

In 2013, the construction of the Zamu Highway, spanning 17 km from Zamu Town in Pome County to Medog County, provided a crucial transportation link. Subsequently, the completion of the Pemo Highway on 6 May 2021, marked the establishment of the final segment in the circular road network centered around Medog County. The Pemo Highway serves as a vital catalyst for the economic and social progress of Medog County and plays a key role in facilitating border defense in TP.

The Pemo Highway is divided into four sections. The first section commences from Pai Town in Milin County and traverses the Duoshungla Mountain. The second section extends from the exit of the Duoshungla Tunnel to Karmajia, followed by the third section from Karmajia to Ani Bridge. The final section spans from Ani Bridge to Jiefang Bridge, totaling a length of 66.86 km. Figure 1 illustrates the geographical location and overview of the Pemo Highway.

The Pemo Highway, situated in the Duoshungla River basin, traverses a region that is predominantly inaccessible and closed to traffic, with a substantial elevation drop of 2892 m from its highest to lowest point. Due to the combination of intense rainfall, heavy snowfall, and various climatic factors, the Pemo Highway has experienced a high occurrence of geological hazards, including landslides, debris flows, rockfalls, and avalanches since its completion. Particularly during the winter season, significant avalanche disasters have frequently transpired, resulting in numerous accidents. Therefore, conducting comprehensive studies on avalanches along the Pemo Highway holds immense importance in fostering local economic and societal development while enhancing the overall production and living standards of the residents.

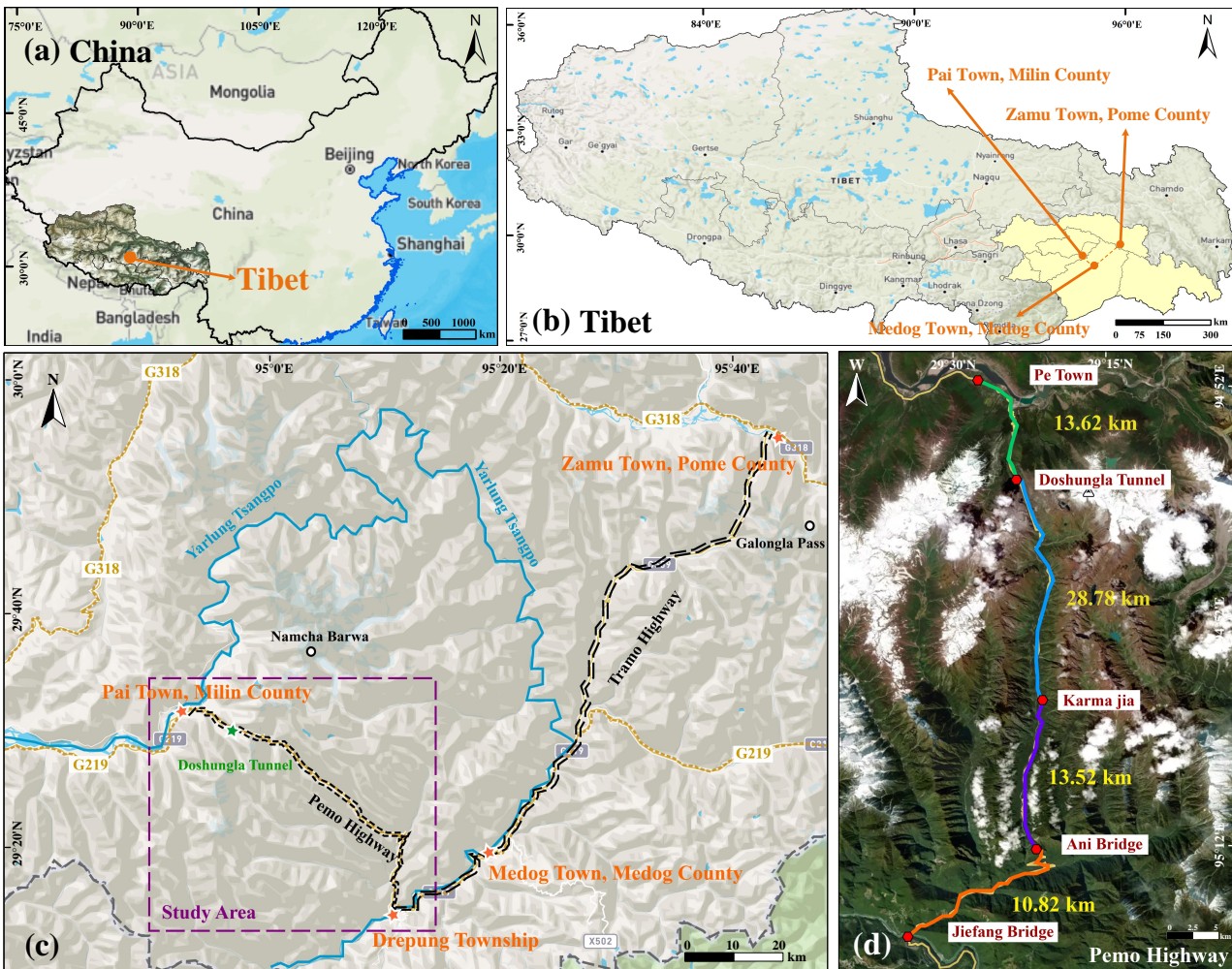

**Figure 1.** Overview of the study area. (**a**) Location of China. (**b**) Location of Tibet. (**c**) Location of the Pemo Highway. (**d**) Overview of the Pemo Highway.

### 2.2. Data Sources

The data used in this paper include Sentinel-2 L1C satellite data [31], digital elevation model (DEM) data, and the climate hazards group infrared precipitation with station (CHIRPS) data [32], all of which are obtained from the GEE dataset. In this paper, the Sentinel-2 data are used to obtain snow cover information, and the DEM data and the CHIRPS rainfall data are used as data sources for the impact factors of avalanches.

Sentinel-2 is a high-resolution multispectral imaging satellite, and Sentinel-2 L1C is geometrically corrected and atmospherically corrected top-of-atmosphere reflectance data. The Sentinel-2 imagery contains 13 spectral bands, and the bands used in this paper include B2 (blue), B3 (green), B4 (red), B8 (NIR), B11 (SWIR1), and B12 (SWIR2). The DEM data are the SRTM 30 m spatial resolution data provided by the National Aeronautics and Space Administration (NASA). The CHIRPS data are global rainfall data from 1981 to the present, which is a gridded time series of rainfall data created by combining 0.05°-resolution satellite imagery with site data.

This paper filters the Sentinel-2 L1C data with less than 10% cloud cover between 16 May 2021 and 30 March 2023 to investigate the spatial and temporal characteristics of the snow cover along the Pemo Highway. The study area in this paper covers both Tile46RFT and Tile46RGT images, and we obtained a total of twelve images after mosaicking the two images at the same moment. The cloud coverage information of the twelve filtered images is shown in Figure 2.

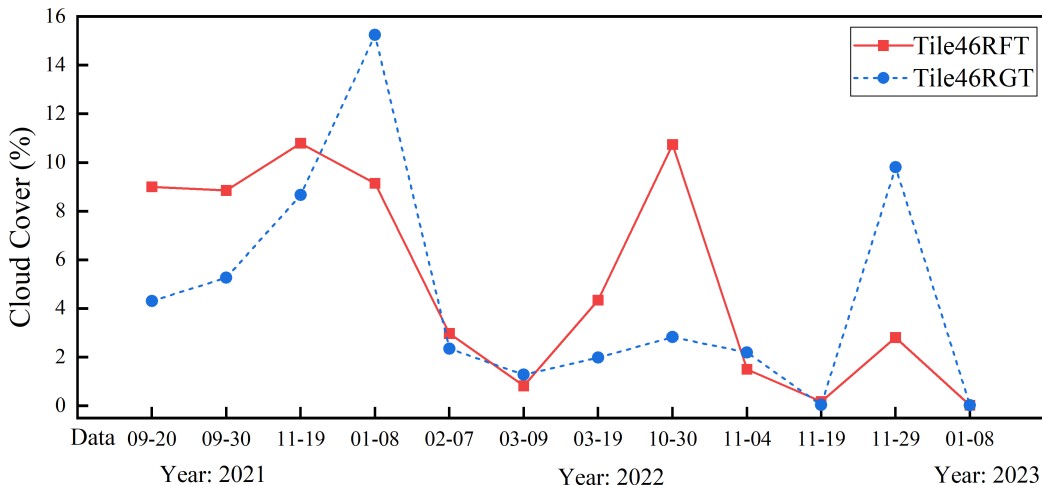

**Figure 2.** Cloud coverage of the used Sentinel-L1C images.

## 3. Spatiotemporal Characteristics of Snow Cover along the Pemo Highway

### 3.1. Snow Identification Method from Sentinel-2 Remote Sensing Data

The five bands of Sentinel-2 satellite data, band 2, band 3, band 4, band 8, and band 11, have better cloud–snow differentiation. The NDSI is very effective in distinguishing between reflectance between the green and NIR bands of the Sentinel-2 image on a pixel-by-pixel basis.

The Sen2Cor tool [33], available from the European Space Agency (ESA), offers a snow detection function that performs atmospheric, topographic, and cirrus corrections on the Sentinel-2 L1C data. Based on this, this paper firstly corrected the twelve selected Sentinel-2 images for atmosphere and topography using the Sen2Cor method and then identified the snow cover elements from the Sentinel-2 images using a combination of NDSI, band spectral thresholding, and band ratio operations, as shown in Equations (1) and (2).

$$NDSI = (Green - SWIR)/(Green + SWIR) \tag{1}$$

$$
\begin{aligned}
NDSI &> 0.2 \\
\rho_{NIR} &> 0.15 \\
\rho_{Blue} &> 0.28 \\
\rho_{Blue}/\rho_{Red} &> 0.85
\end{aligned}
\tag{2}
$$

### 3.2. Investigated Spatial and Temporal Characteristics of Snow Cover along the Pemo Highway

This paper investigated the characteristics of snow cover changes along the Pemo Highway based on the above methods, and the results are shown in Figure 3. It can be concluded that the spatial and temporal variations in snow cover along the Pemo Highway have the following characteristics:

(1) The duration of snow cover in the study area is relatively long and has an annual periodicity, with snow cover on the ground mainly distributed between late October and early April of the following year, with a period of about five months.

(2) The snow cover along the Pemo Highway is mainly located at the Duoshungla Tunnel. Around late October, the snow cover first appears at the Duoshungla Tunnel, then gradually extends along the Duoshungla Tunnel towards Lager until March, when the snow and ice start to melt and the snow cover decreases.

In summary, the main snow-covered section along the Pemo Highway is the Duoshungla Tunnel to Lager, with the remaining three sections having less snow cover. From January to February each year, the snow cover on the Duoshungla Tunnel section is the largest.

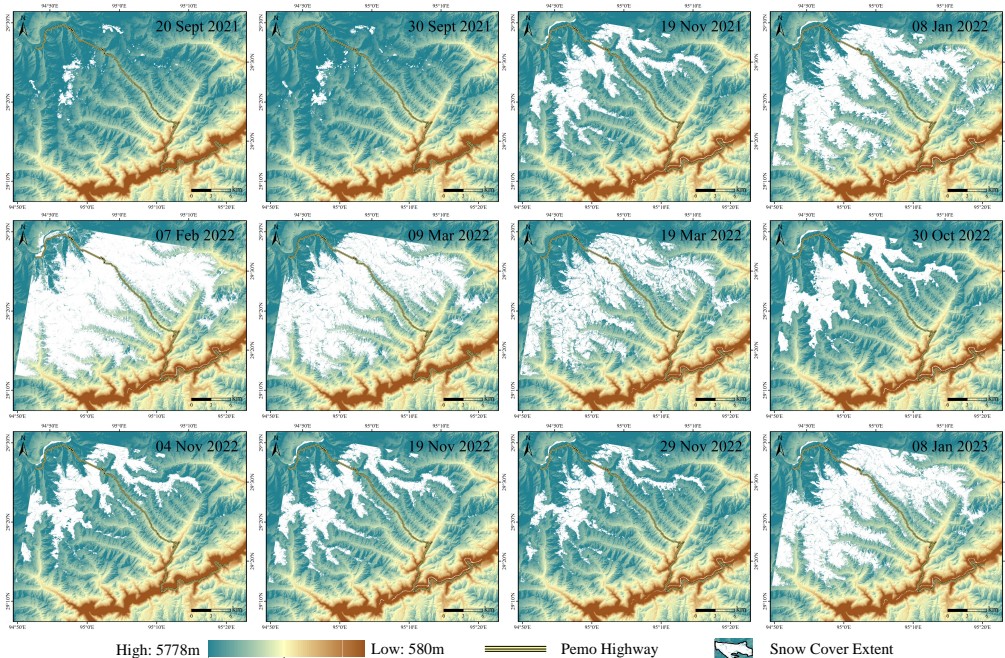

**Figure 3.** Results of snow cover extraction in the study area.

By comparing the snow identification results with the satellite imagery, we concluded that the identified snow elements obviously excluded dense vegetation, water bodies, and thin clouds. However, it was not possible to distinguish between glaciers, frozen rivers, and snow, as shown in Figure 4.

(1) The identification results cannot distinguish frozen rivers. For example, in September 2021 and March 2022, the waters of the Yarlung Tsangpo River in the northeastern region of the study area were not identified, while from November 2022 to January of the following year, weather changes caused the river surface of part of the Yarlung Tsangpo River to be frozen, which was identified as snow, as shown in Figure 4a.

(2) The identification results cannot distinguish glaciers, for example, the glaciers located on the Duoshungla Mountain and the Namcha Barwa, as shown in Figure 4b.

(3) The identification results accurately distinguish between thin clouds and snow, as shown in Figure 4c.

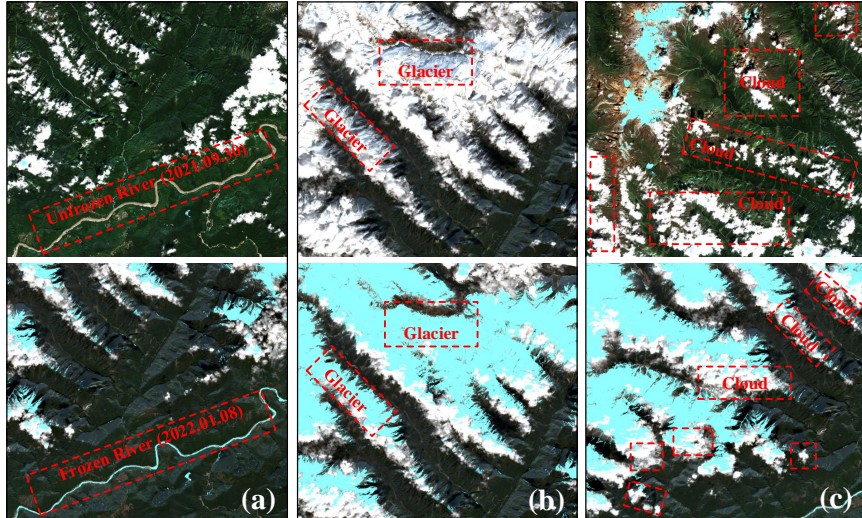

**Figure 4.** Comparison of snow identification results with satellite images. (**a**) Identification accuracy of a river. (**b**) Identification accuracy of a glacier. (**c**) Identification accuracy of thin clouds.

## 4. Avalanche Susceptibility Assessment along the Pemo Highway

We concluded that the period of snow cover along the Pemo Highway is from November to March each winter, mainly in the section from the Duoshungla Tunnel to Lager. Historical studies have shown that avalanches along the Pemo Highway begin to occur around mid-January when the snow accumulates to a certain thickness and the weather warms up. Avalanche hazards result from interactions between snow, climatic, topographic, and subsurface factors.

This paper first presented the NDSI as a major condition factor of avalanche, replacing the factor of snow depth in previous studies, and established a total of six condition factors, including the NDSI, altitude, slope, mean monthly precipitation, vegetation cover, and vegetation type. Second, we determined the weights of the condition factors based on the analytic hierarchy process (AHP) method. Finally, we mapped the avalanche susceptibility zones along the Pemo Highway for January, February, and March 2023 and discussed the avalanche hazard risk for each section of the highway. The workflow of the avalanche susceptibility assessment along the Pemo Highway is shown in Figure 5.

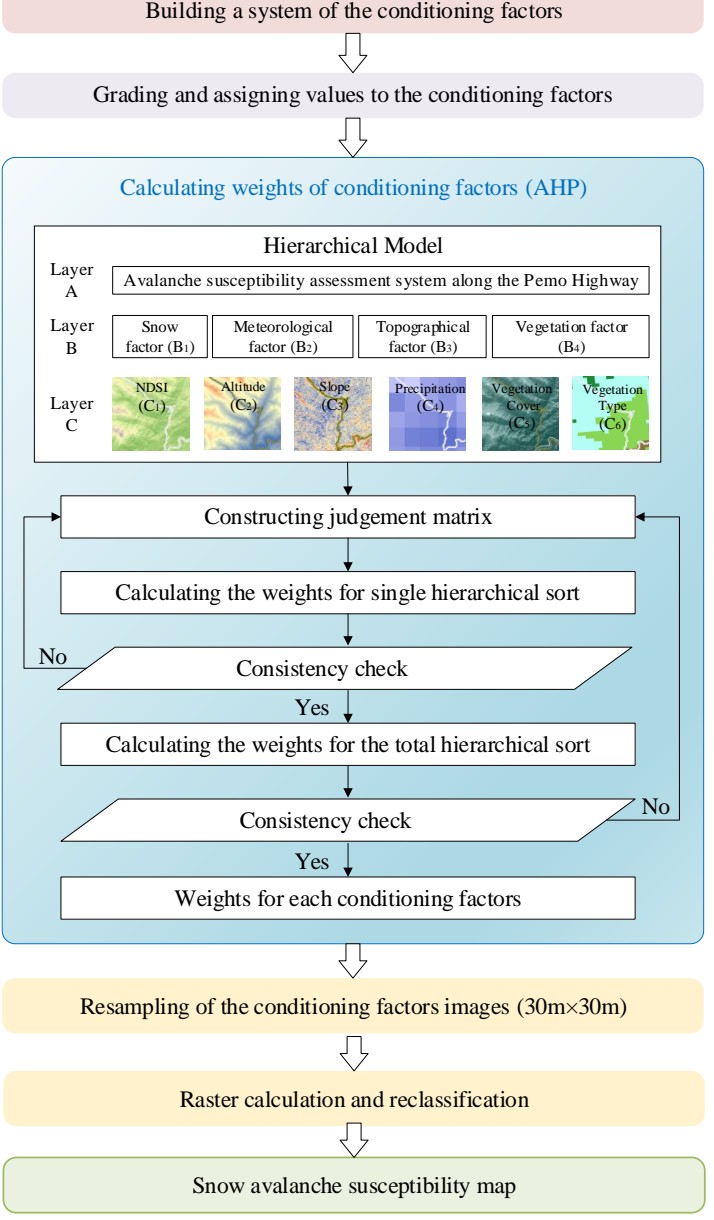

**Figure 5.** The workflow of avalanche susceptibility assessment.

### 4.1. Extraction of Conditioning Factors for Avalanches

This paper develops six avalanche conditioning factors in terms of snow, topography, meteorology, and vegetation. In fact, the three meteorological factors of temperature, wind speed, and precipitation all influence the occurrence of avalanches [30]. However, due to the fact that the study area is small, the differences in temperature and wind speed are generally consistent across the study area, while there are differences in precipitation. For this reason, only the precipitation factor was considered in the mapping of the avalanche susceptibility. The following is a summary of the avalanche impact factors in the study area. The overview of each avalanche condition factor along the Pemo Highway is shown below.

(1) Snow factor—NDSI

The maximum value of NDSI for January 2023 along the Pemo Highway was 0.68; the maximum value of NDSI for February was 0.53; and the maximum value of NDSI for March was 0.58. The monthly average value of NDSI for January, February, and March 2023 is shown in Figure 6a.

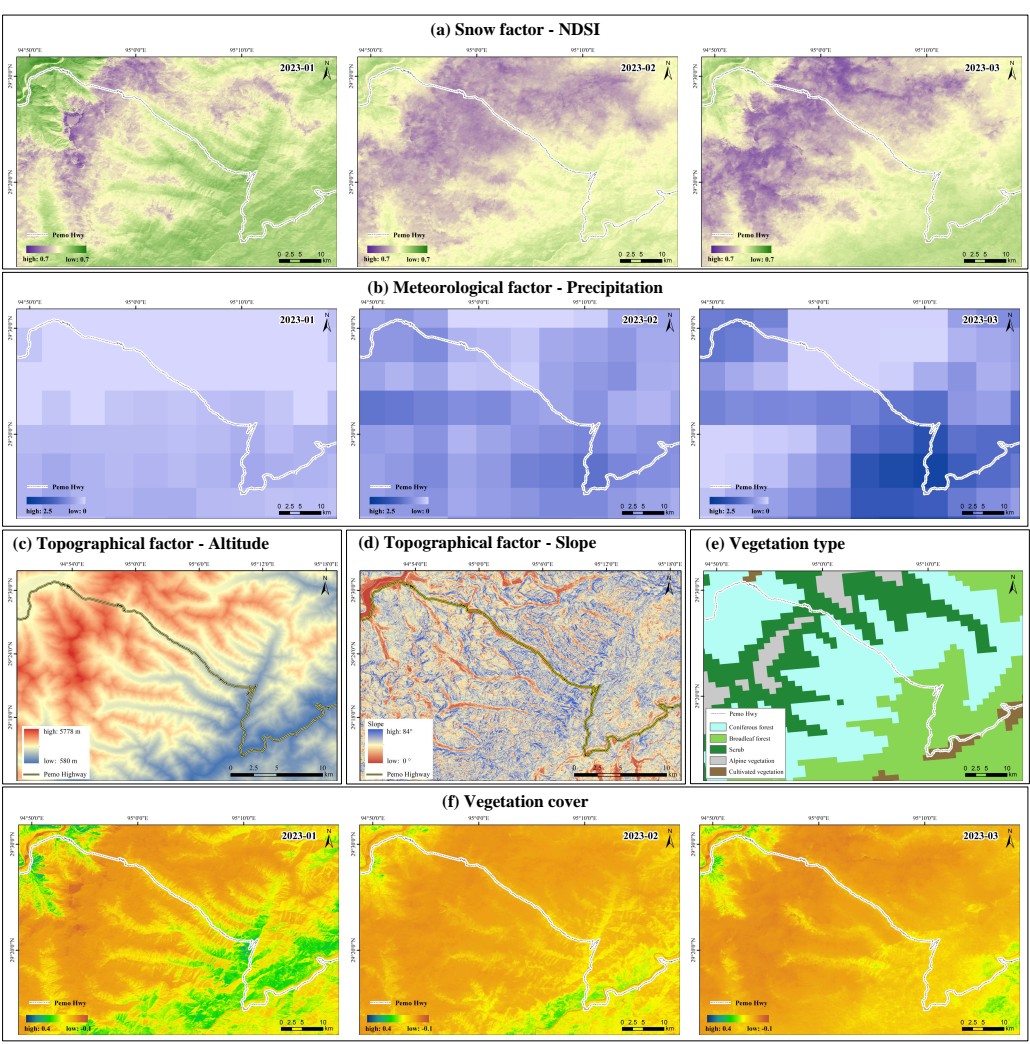

**Figure 6.** The conditioning factors for avalanche susceptibility.

(2) Topographical factors

In this paper, altitude and slope are chosen as topographic condition factors. As shown in Figure 6c,d, the highest altitude along the Pemo Highway is 5779 m, and the lowest is 580 m; the maximum slope is 84°.

(3) Meteorological factors

The study area had a minimum temperature of $-12$ °C and a maximum temperature of $-2$ °C in January, a minimum temperature of $-9$ °C and a maximum temperature of $0$ °C in February, and a minimum temperature of $-4$ °C and a maximum temperature of $5$ °C in March. The maximum wind speed in the study area was 1.2 m/s in January, 0.7 m/s in February, and 0.76 m/s in March. The maximum precipitation in the study area was 3.3 mm in January, 7 mm in February, and 12.8 mm in March. The monthly average precipitation data for the study area from January to March 2023 are shown in Figure 6b.

(4) Vegetation factor

This paper calculated the vegetation cover of the study area based on band 4 and band 8 of the Sentinel-2 data, as shown in Equation (3). As shown in Figure 6f, the highest vegetation cover along the Pemo Highway in 2023 was 0.28 in January; 0.22 in February; and 0.37 in March.

$$NDVI = (NIR - RED)/(NIR + RED) \tag{3}$$

The vegetation type data in this paper were obtained from the 1:1 million vegetation type spatial distribution dataset in China. The vegetation cover types in the study area are shown in Figure 6e. There are five vegetation types along the Pemo Highway: coniferous forest, broad-leaved forest, scrub, alpine vegetation, and cultivated vegetation, of which the main vegetation type is coniferous forest.

### 4.2. Determination of the Weights of the Conditioning Factors

Before determining the weights of the conditioning factors, it is necessary to classify the avalanche condition factors into different levels of importance and assign values to them. Based on the actual situation in the study area and the experience of relevant references, the values of the avalanche susceptibility condition factors constructed in this paper are shown in Table 1.

**Table 1.** Graded assignment of conditioning factors for avalanche susceptibility.

| Primary Factor | Secondary Factor | Factor Importance Assignment |
| --- | --- | --- |
| Snow | NDSI | NDSI $\leq$ 0, value = 0; 0 < NDSI $\leq$ 0.2, value = 1; 0.2 < NDSI $\leq$ 0.4, value = 4; 0.4 < NDSI $\leq$ 0.6, value = 8; 0.6 < NDSI $\leq$ 0.8, value = 10. |
| Topography | Slope | 0° < slope $\leq$ 15° or slope $\geq$ 70°, value = 2; 15° < slope $\leq$ 25° or 60° < slope < 70°, value = 4; 45° < slope < 60°, value = 6; 25° < or < 35°, value = 8; 35° $\leq$ slope $\leq$ 45°, value = 10. |
| | Altitude | altitude $\leq$ 1000 m, value = 2; 1000 m < altitude $\leq$ 2000 m, value = 4; 2000 m < altitude $\leq$ 3000 m, value = 6; 3000 m < altitude $\leq$ 4000 m or altitude > 5000 m, value = 8; 4000 m < altitude $\leq$ 5000 m, value = 10. |
| Meteorology | Average Monthly Precipitation | 0 mm $\leq$ 3 mm, value = 2; 3 mm < AMC $\leq$ 6 mm, value = 4; 6 mm < AMC $\leq$ 8 mm, value = 6; 8 mm < AMC $\leq$ 10 mm, value = 8; AMC >10 mm, value = 10. |
| Vegetation | Vegetation Cover | NDVI $\leq$ 0.1 or 0.6 < NDVI $\leq$ 0.8, value = 4; 0.1 < NDVI $\leq$ 0.2, value = 10; 0.2 < NDVI $\leq$ 0.4, value = 8; 0.4 < NDSI $\leq$ 0.6, value = 6; NDVI > 0.8, value = 2. |
| | Vegetation Type | Broad-leaved forest = 1; Coniferous forest = 2; Scrub = 5; Alpine vegetation = 6; Cultivated vegetation = 4. |

This paper determines the weights of each condition factor based on the AHP method [20,34], and the specific steps are as follows:

(1) Constructing the hierarchical structure model. The hierarchical structure model of the study area constructed in this paper is shown in Figure 5, which is divided into three main layers: target layer (A), criterion layer (B), and indicator layer (C).

(2) Constructing the judgment matrix A for each layer. The judgment matrix between the upper and lower layers describes the correlation among the conditioning factors. In this paper, we constructed a judgment matrix based on a 1–9 scale method from Thomas L. Saaty. A = $(a_{ij})_{n*n}$, where $a_i$, $a_j$ are condition factors, $a_{ij}$ is the matrix composed of all

condition factors, and $n$ is the matrix order. The results of the established judgment matrix are shown in Tables 2–4.

**Table 2.** The judgment matrix of layer B.

| Value | $B_1$ | $B_2$ | $B_3$ | $B_4$ |
|---|---|---|---|---|
| $B_1$ | 1 | 1 | 2 | 3 |
| $B_2$ | 1 | 1 | 2 | 3 |
| $B_3$ | 1/2 | 1/2 | 1 | 2 |
| $B_4$ | 1/3 | 1/3 | 1/2 | 1 |

**Table 3.** The judgment matrix of topographic factors ($B_2$).

| Value | $C_2$ | $C_3$ |
|---|---|---|
| $C_2$ | 1 | 2 |
| $C_3$ | 1/2 | 1 |

**Table 4.** The judgment matrix of vegetation factors ($B_4$).

| Value | $C_5$ | $C_6$ |
|---|---|---|
| $C_5$ | 1 | 2 |
| $C_6$ | 1/2 | 1 |

(3) Calculating the matrix for each layer and performing a consistency test. To ensure that the weights obtained from the judgment matrix are available, a consistency test should be performed on them. The test formula is $CR = CI/RI$, $CI = (\lambda_{max} - n)/(n - 1)$; where CR and RI are random consistency indicators, CI is the consistency indicator, $\lambda_{max}$ is the maximum eigenvalue of the judgment matrix, and $n$ is the order of the judgment matrix.

(4) Calculating the overall matrix and performing a consistency test. The final calculated weight for each avalanche condition factor in this paper is shown in Table 5.

**Table 5.** Calculated weights of condition factors for avalanche susceptibility.

| Target | Criterion | $W(A/B_j)$ | Index | $W(B_j/C_j)$ | $W(A_j/C_j)$ |
|---|---|---|---|---|---|
| | Snow factor ($B_1$) | 0.3509 | NDSI ($C_1$) | 1 | 0.3509 |
| | Topographic factors ($B_2$) | 0.3509 | Altitude ($C_2$) | 0.6667 | 0.2339 |
| | | | Slope ($C_3$) | 0.3333 | 0.1170 |
| Avalanche susceptibility | Meteorological factor ($B_3$) | 0.1891 | Precipitation ($C_4$) | 1 | 0.1891 |
| | Vegetation factors ($B_4$) | 0.1091 | Vegetation cover ($C_5$) | 0.6667 | 0.0727 |
| | | | Vegetation type ($C_6$) | 0.3333 | 0.0364 |

After determining the weights of the condition factors, this paper resampled the raster images of the six condition factors consistently to a spatial resolution of 30 m × 30 m and then performed raster calculation and reclassification to map the final avalanche susceptibility zones of the study area. Finally, this paper classifies avalanche susceptibility based on the avalanche risk index (ARI) [20,22,35], which is the sum of the weight and the graded value of all condition factors. As a result, the study area is divided into five parts: high-susceptibility areas, medium-susceptibility areas, low-susceptibility areas, and very low susceptibility areas.

*4.3. Results of Avalanche Susceptibility Mapping along the Pemo Highway*

In this paper, the results of avalanche susceptibility mapping along the Pemo Highway for January, February, and March 2023 are shown in Figure 7. The lengths of roads within the predicted various susceptibility zones are listed in Table 6. Significantly, the recent "1.17 avalanche", which is discussed in detail in Section 5.1, is located within the very

high avalanche susceptibility zones for January 2023 predicted in this paper, as shown in Figure 7a.

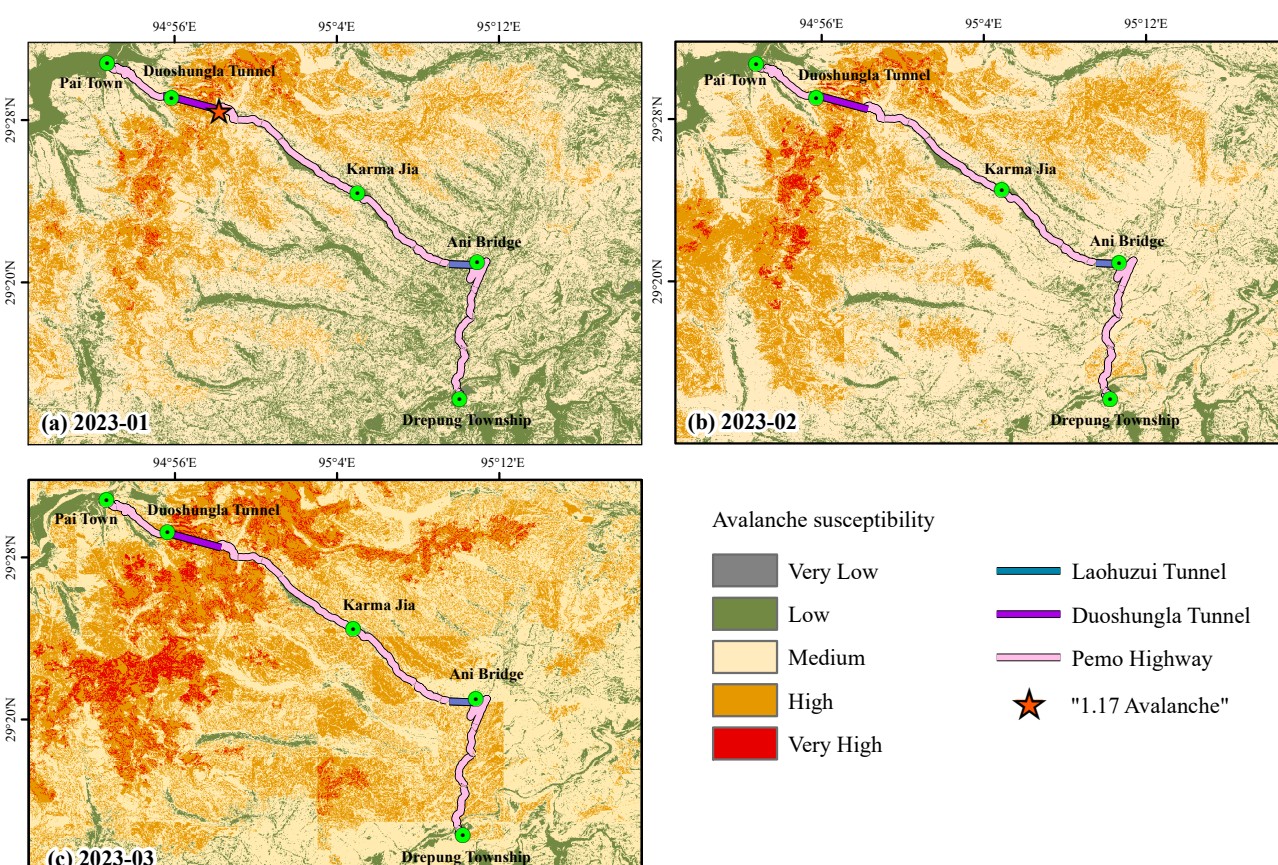

**Figure 7.** Avalanche susceptibility maps along the Pemo Highway. (**a**) Avalanche susceptibility map for January 2023. (**b**) Avalanche susceptibility map for February 2023. (**c**) Avalanche susceptibility map for March 2023.

**Table 6.** The lengths of roads within the predicted various avalanche susceptibility zones.

| Length (km) | Very High | High | Medium | Low | Very Low |
|---|---|---|---|---|---|
| January 2023 | 0.617 | 0.483 | 28.309 | 32.517 | 0.614 |
| February 2023 | 1.788 | 5.944 | 29.167 | 29.961 | 0 |
| March 2023 | 4.731 | 15.897 | 31.322 | 14.910 | 0 |

In summary, the results of the avalanche susceptibility predicted in this paper reveal that: (1) Avalanches are more likely to occur within the slope range of 35° to 45° and at elevations between 3500 m and 4000 m. (2) The very high susceptibility areas along the Pemo Highway are mainly located along the road from Duoshungla Tunnel to Lager, and the low-susceptibility areas are mainly located along the roads from Karma Jia to Ani Bridge. (3) The risk of avalanches along the Pemo Highway experiences a notable increase from January to March, with March identified as the month when avalanches are most likely to occur.

The Pemo Highway includes two critical projects, the Duoshungla Tunnel and the Laohuzui Tunnel. In winter, the tunnel exits are prone to avalanches due to high wind speeds. This paper provides a detailed analysis of the geographic situation and avalanche susceptibility at the exits of the Duoshungla Tunnel and the Laohuzui Tunnel in particular.

The Duoshungla Tunnel is located at the beginning of the second road of the Pemo Highway, with a total length of 4.78 km and a maximum burial depth of 820 m. The tunnel

entrance is situated at an altitude of 3547 m with a slope of approximately 25°, while the exit is at an altitude of 3566 m with a slope of approximately 40°. The Laohuzui Tunnel is situated at the end of the third section of the Pemo Highway, with a total length of 1.92 km. The tunnel entrance is at an altitude of 1918 m with a slope of approximately 42°, and the exit is at an altitude of 1615 m with a slope of approximately 20°.

According to the results of avalanche susceptibility predicted in this paper, avalanches are prone to occur in areas with slopes from 35° to 45° and at elevations between 3500 m and 4500 m. Additionally, the occurrences of transverse wind at the tunnel exits can further increase avalanche susceptibility. It can be inferred that vehicles face an extremely high avalanche risk at the exit of the Duoshungla Tunnel, whereas the risk at the exit of the Laohuzui Tunnel is relatively low.

From Figure 7c, particularly during March, there exists a 3.7 km long stretch of road with very high avalanche risk from the exit of the Duoshungla Tunnel in the direction towards Karma Jia. Based on these findings, we propose the following recommendations: (1) Implement traffic control and timely warnings for tourists and local residents during March each year to prevent casualties; (2) during the snow season, employ relevant engineering measures, such as snow barriers, along the road from the exit of the Duoshungla Tunnel to Karma Jia.

## 5. Discussion

### 5.1. Avalanche at the Exit of the Duoshungla Tunnel on 17 January 2023

On 17 January 2023, a significant avalanche event took place at the exit of the Duoshungla Tunnel on the Pemo Highway [2]. The exit of the Duoshungla Tunnel is located on the second road of the Pemo Highway in the direction from Lager to Medog, shown in Figure 8a, and the topographic situation around the exit of the tunnel is shown in Figure 8b. As evident in Figure 8d, the avalanche caused heavy casualties and attracted the concern of the local government.

Eyewitness accounts from the rescued individuals described the avalanche as exceptionally severe, with large masses of snow descending rapidly, resulting in cars being overturned on the spot. By January 20, numerous vehicles were buried under the snow, and the avalanche caused injuries to 28 people. Over the past three years, a series of avalanches have occurred near the exit road of the Duoshungla Tunnel, including a fatal avalanche incident on 1 April 2021, which claimed four lives, and another avalanche event on 5 February 2022, that resulted in five individuals being buried.

The exit of the Duoshungla Tunnel, positioned at an altitude of 3566 m and featuring a slope of approximately 31.4°, has exhibited a propensity for avalanche occurrences in previous years due to its unique geographical location. Experienced local drivers who regularly travel between Medog County and Pai Town have noted that this avalanche transpired earlier than in previous years, as avalanches generally transpired during warmer weather conditions in recent years. Remarkably, this avalanche event represents the first occurrence in January.

The location of this avalanche incident aligns with the high avalanche susceptibility zone identified in this paper for January 2023, providing evidence of the feasibility and accuracy of the research conducted. Specifically, the exit of the Duoshungla Tunnel (Figure 8b) and the adjacent exit road (Figure 8c) are recognized as the most hazardous sections along the Pemo Highway in terms of avalanche occurrences. Consequently, it is imperative to enhance and implement robust avalanche prevention and control measures for these specific areas.

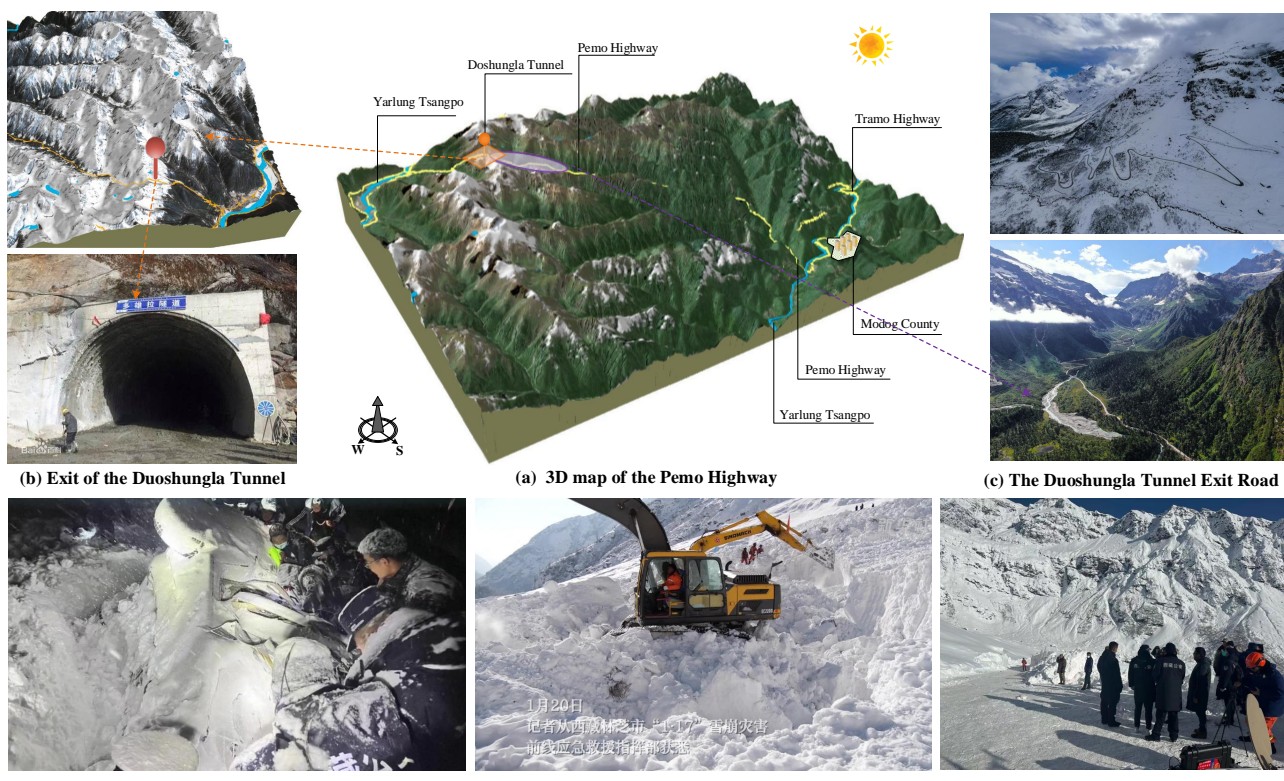

**Figure 8.** Overview of the Duoshungla Tunnel and the "1.17 avalanche". (**a**) 3D map of the Pemo Highway. (**b**) Exit of the Duoshungla Tunnel. (**c**) The Duoshungla Tunnel exit road. (**d**) Overview of the "1.17 avalanche".

*5.2. Contributions to the Community*

The "1.17 avalanche" at the Duoshungla Tunnel resulted in casualties among both local residents and foreign tourists, highlighting the lack of awareness regarding the avalanche risk along the Pemo Highway. Notably, the Pemo Highway was fully completed, including the hardening of the road surface, in October 2022, making it less than a year since its completion. However, this year has witnessed various hazards along the highway, and only a limited number of researchers have conducted comprehensive investigations and studies in this area. Given that the Pemo Highway serves as a vital transportation route critical to the economic development of Medog County, it is imperative to address the increased risk of avalanches due to rising global temperatures [36].

Our research endeavors to contribute to a heightened understanding of the Pemo Highway by encouraging further investigations in this field. Additionally, we aim to raise awareness among local residents about the potential hazards posed by avalanches, ensuring that they are well-informed and prepared to mitigate these risks effectively. Through our work, we aspire to foster a timely response and enhanced vigilance within the community regarding avalanche hazards along the Pemo Highway.

*5.3. Limitations and Future Studies*

5.3.1. Limitations of Available Data in the Study Area

On the one hand, the Pemo Highway is located in a mountainous area, which poses difficulties in setting up various monitoring stations along the highway; on the other hand, it has been two years since the Pemo Highway was first opened to traffic in 2021, and few researchers have focused their attention here, in addition to the fact that there are still few studies on avalanche hazards in China. As a result, there is a lack of available historical avalanche statistical data for the study area covered by the Pemo Highway. In this paper,

the analysis of avalanche susceptibility along the Pemo Highway is limited to the literature from other similar areas to obtain the weight of each condition factor.

### 5.3.2. Limitation of Analysis Method for Avalanche Susceptibility

The paper establishes the weights of conditioning factors for avalanche susceptibility in the study area using the AHP method. However, it is essential to acknowledge that while the AHP method proves effective and feasible in the absence of training data, it necessitates subjective reclassification of conditioning factors and assigning weights, posing challenges in accurately describing the weighted relationships between avalanche disasters and non-linear conditioning factors. In contrast, machine-learning-based methods offer several advantages, including rapid training speed and the ability to capture intricate nonlinear relationships between avalanches and triggering factors, owing to their capacity to leverage substantial amounts of training data. Notably, decision-tree-based models [37], when used for quantifying influencing factors, can even provide a certain degree of interpretability.

Regrettably, the study encounters limitations due to the unavailability of historical avalanche data in the study area, which restricts the researchers to performing an avalanche susceptibility analysis using the AHP method. Nevertheless, the application of the AHP method provides valuable insights, given the constraints in obtaining extensive training data for machine learning models.

### 5.3.3. Outlook

The Medog County-centered circular highway is the main channel of communication between Medog and the outside world, as well as being crucial to the economic construction of Medog County. In our future research work, we will expand the scope of the study area to investigate the avalanche hazard risk of the entire circular highway centered on Medog, including the Pemo Highway, the Zamu Highway, and the G318 National Highway, and explore the application potential of deep learning in avalanche susceptibility based on the existing data to better predict the potential avalanche hazards along the highway.

### 6. Conclusions

In this paper, we first investigate the spatiotemporal characteristics of snow cover along the Pemo Highway over the past two years based on the GEE platform. Second, we assess the avalanche susceptibility along the Pemo Highway in January, February, and March 2023 by considering snow, topography, meteorology, and subsurface factors using the AHP method. The results of snow cover identification based on Sentinel-2 data indicate that snow cover along the Pemo Highway persists from November to March each winter, predominantly occurring on the road between the Duoshungla Tunnel and Lager. The results of the avalanche susceptibility assessment reveal that the exit of the Duoshungla Tunnel is particularly prone to avalanches during winter, especially in March, with an avalanche risk present along a stretch of approximately 3.7 km from the tunnel exit to Lager. Finally, we discussed the recent "1.17 avalanche" event along the Pemo Highway, which demonstrates the accuracy of the avalanche susceptibility analysis presented in this paper. The research findings can serve as a reference for implementing traffic control or avalanche prevention measures along the Pemo Highway to reduce casualties.

**Author Contributions:** Conceptualization, N.X. and G.M.; methodology, N.X. and G.M.; software, N.X.; investigation, N.X. and G.M.; data curation, N.X. and G.M.; writing—original draft preparation, N.X.; writing—review and editing, N.X. and G.M.; funding acquisition, G.M. All authors have read and agreed to the published version of the manuscript.

**Funding:** This research was jointly supported by the National Natural Science Foundation of China (Grant No. 42277161), and the China University of Geosciences (Beijing) Postgraduate Innovation Grant Programme (ZD2023YC049).

**Data Availability Statement:** Not applicable.

**Acknowledgments:** The authors would like to thank the editor and the reviewers for their contributions.

**Conflicts of Interest:** The authors declare no conflict of interest.

**Abbreviations**

The following abbreviations are used in this manuscript:

| | |
|---|---|
| AHP | Analytic hierarchy process |
| CART | Classification and regression tree |
| CHIRPS | Climate hazards group infrared precipitation with station |
| DEM | Digital elevation model |
| ESA | European Space Agency |
| GEE | Google Earth Engine |
| KNN | K-nearest neighbor neighbor |
| RF | Random forest |
| SVM | Support vector machine |
| TP | Tibet Plateau |
| MCDA | Multi-criteria decision analysis |
| NDSI | Normalized difference snow index |
| NDVI | Normalized difference vegetation index |
| NASA | National Aeronautics and Space Administration |

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
