# Peer review of "Avalanche Susceptibility Mapping by Investigating Spatiotemporal Characteristics of Snow Cover Based on Remote Sensing Imagery along the Pemo Highway—A Critical Transportation Road in Tibet, China"

_water, doi:10.3390/w15152743_

Round 1
Reviewer 1 Report
The manuscript has been thoroughly prepared and is appropriate for the readership of the journal. It delves into the generation of a snow avalanche susceptibility map using six conditioning factors and the Analytic Hierarchy Process (AHP) method. Notably, this study does not utilize avalanche training data but instead employs Normalized Difference Snow Index (NDSI) data derived from Sentinel-2 images to assess snow cover occurrence in the study area. Although this approach is not groundbreaking, it can be deemed acceptable under data-scarce conditions. Consequently, the paper can be seen as a commendable implementation of well-established methodologies. However, I have some concerns regarding resolution consistency, accuracy assessment, literature review, and the discussion section. Additionally, there is a need for textual and artistic editing in the manuscript. Here are my comments:
MAJOR COMMENTS:
1) The spatial resolution of the conditioning factors appears to be inconsistent, and it seems that resampling was not applied. Typically, all conditioning factors should be resampled to a final resolution that matches the resolution of the susceptibility maps. It would be beneficial to elucidate to the reviewers the rationale behind not resampling these factors.
2) It is necessary to highlight why the AHP method was chosen over machine learning methods. While I understand that the lack of snow avalanche samples led to the implementation of the AHP method for ranking the conditioning factors, it is important to note that the field tends to lean towards the use of data-driven techniques. Therefore, it would be appropriate to mention studies employing both AHP and machine learning methods (not solely CNN-based models) in the introduction, along with their respective advantages and disadvantages. Additionally, in the discussion section, you should compare the limitations and advantages of the AHP approach with those of ML-based studies. I have provided a list of recent papers that utilize ML-based solutions for snow avalanche susceptibility, which you may find useful. Interestingly, your literature review lacks a comprehensive examination of snow avalanche susceptibility studies, instead relying on studies pertaining to other types of geohazards to support your ideas.
3) The absence of validation data hinders the ability to generate accuracy assessments for your maps, as well as the distribution of susceptibility classes for known past avalanches. Undeniably, this constitutes a significant limitation in your research. To address this limitation, it is crucial to include the position of the 1.17 Avalanche near Duoshungla Tunnel on your susceptibility map. Furthermore, providing numerical output to showcase the susceptibility of that specific avalanche zone would significantly enhance the credibility of your maps.
4) Please provide multicollinearity test results or correlation coefficients for all of the conditioning factors.
MINOR COMMENTS:
5) Figures 1 and 4 include annotated sub-figures labeled as (a), (b), (c), and (d). These sub-figures should also be explained in the figure captions. Additionally, the maps would benefit from the inclusion of cartographic elements such as a scale and a north arrow.
6) In Figure 2, please remove the seconds from the grid coordinates, and provide a legend explaining the colours presented in the maps. Moreover, revise the dates to be clearer, for example, using a format like "20 Sept 2021."
7) Please cite local or international resources that support the occurrences of geohazards in your study area, as mentioned in lines 105-111.
8) In Equations 1 and 3, please provide the names of the bands rather than their numbers, as band numbers can vary across different sensors.
9) Please provide references or resources explaining the rationale behind your selection of the Avalanche Risk Index (ARI) to generate susceptibility classes, instead of other methods such as quartiles and natural breaks (as mentioned in lines 243-246).
RESOURCES:
https://doi.org/10.1007/s00477-023-02392-6
https://doi.org/10.3390/rs14061340
https://doi.org/10.1038/s41598-020-75476-w
Author Response
Dear Reviewer,
We would like to submit our revised paper entitled “Avalanche susceptibility mapping by investigating spatiotemporal characteristics of snow cover based on remote sensing imagery along the Pemo Highway – a critical transportation road in Tibet, China” for your consideration for publication in Water.
We have made a point-by-point response to the reviewer’ comments and suggestions, including a detailed description of any requested or suggested revisions.
We have also carefully checked and corrected the writing format and modified the structure and description of this paper to better fit the style of Water.
All the modifications and explanations in this revised version are listed in detail in the following “Responses to the Reviewers” and “Marked Manuscript”.
We would deeply appreciate your consideration and reviewers’ helpful comments and suggestions.
Yours Sincerely,
Ning Xi, Gang Mei*
School of Engineering and Technology,
China University of Geosciences (Beijing)
Email: [email protected]; [email protected]

Reviewer 2 Report
Pemo Highway is a critical transportation road in Tibet Plateau. In this paper, the authors investigate the spatiotemporal characteristics of snow cover and assess the avalanche risk along the Pemo Highway in time. A recent avalanche event validates the accuracy of the research work. The paper is instructive for the construction of Medog County. Several problems are suggested to be revised as follows.
1. Page 2, Line 57. The authors are suggested to update several recent references in Introduction about the applications of GEE.
2. Page 2, Line 64. Here the authors describe a preference for snow depth as an important condition factor of avalanche susceptibility in past studies. Some recent relevant references about the historical studies should be included.
3. In Figure 2, the authors describe the cloud coverage of the remote sensing images used in the paper; however, why are there two curves in the figure? Please explain.
4. Please add a legend to Figure 3 to clearly show the content of the figure.
5. Section 5.1. Add a description of Figure 8(a) to the text.
6. Page 10, Line 260. Please add a more detailed description of the Duoshungla Tunnel and the Laohuzui Tunnel.
7. The paper provides a detailed investigation of snow cover and avalanche susceptibility along the Pamo highway. It is recommended that the practical implications of the findings be further highlighted and more specific recommendations for avalanche prevention provided in Discussion.
8. Please provide the full text of the following abbreviations in Section 2: CHIRPS, NASA, ESA.
Acceptable
Author Response

(The authors gave the same response as above.)

Round 2
Reviewer 1 Report
Many thanks for the revision. Whilst the authors did not furnish a response letter encompassing responses to line-by-line comments, I was able to discern the modifications in the text. I can confidently state that the authors have indeed clarified the issues I raised. At this juncture, I have no further comments to add.